

# Uncertainty modeling process for semantic technology

Rommel N. Carvalho[1,2], Kathryn B. Laskey[3] and Paulo C.G. Da Costa[3]

[1] Department of Research and Strategic Information, Office of the Comptroller General of Brazil, Brasília, DF, Brazil
[2] Department of Computer Science, Universidade de Brasília, Brasília, DF, Brazil
[3] Department of Systems Engineering and Operations Research, George Mason University, Fairfax, VA, United States of America

## ABSTRACT

The ubiquity of uncertainty across application domains generates a need for principled support for uncertainty management in semantically aware systems. A probabilistic ontology provides constructs for representing uncertainty in domain ontologies. While the literature has been growing on formalisms for representing uncertainty in ontologies, there remains little guidance in the knowledge engineering literature for how to design probabilistic ontologies. To address the gap, this paper presents the Uncertainty Modeling Process for Semantic Technology (UMP-ST), a new methodology for modeling probabilistic ontologies. To explain how the methodology works and to verify that it can be applied to different scenarios, this paper describes step-by-step the construction of a proof-of-concept probabilistic ontology. The resulting domain model can be used to support identification of fraud in public procurements in Brazil. While the case study illustrates the development of a probabilistic ontology in the PR-OWL probabilistic ontology language, the methodology is applicable to any ontology formalism that properly integrates uncertainty with domain semantics.

## INTRODUCTION

The ability to represent and reason with uncertainty is important across a wide range of domains. For this reason, there is a need for a well-founded integration of uncertainty representation into ontology languages. In recognition of this need, the past decade has seen a significant increase in formalisms that integrate uncertainty representation into ontology languages. This has given birth to several new languages such as: PR-OWL (*Costa, 2005*; *Costa, Laskey & Laskey, 2005*; *Costa, Laskey & Laskey, 2008*; *Carvalho, 2011*; *Carvalho, Laskey & Costa, 2013*), OntoBayes (*Yang & Calmet, 2005*), BayesOWL (*Ding, Peng & Pan, 2006*), P-CLASSIC (*Koller, Levy & Pfeffer, 1997*) and probabilistic extensions of SHIF(**D**) and SHOIN(**D**) (*Lukasiewicz, 2008*).

However, the increased expressive power of these languages creates new challenges for the ontology designer. In addition to developing a formal representation of entities and relationships in a domain, the ontology engineer must develop a formal characterization of the uncertainty associated with attributes of entities and relationships among them.

Corresponding author
Kathryn B. Laskey, klaskey@gmu.edu

While a robust literature exists in both ontology engineering (*Allemang & Hendler, 2011*; *Gomez-Perez, Corcho & Fernandez-Lopez, 2004*) and knowledge engineering for probability models (*Laskey & Mahoney, 2000*; *Druzdzel & Van der Gaag, 2000*; *Korb & Nicholson, 2003*; *O'Hagan et al., 2006*), these fields have developed largely independently. The literature contains very little guidance on how to build ontologies that capture knowledge about domain uncertainties.

To fill the gap, this paper describes the Uncertainty Modeling Process for Semantic Technology (UMP-ST), a methodology for defining a probabilistic ontology and using it for plausible reasoning in applications that use semantic technology. The methodology is illustrated through a use case in which semantic technology is applied to the problem of identifying fraud in public procurement in Brazil. The purpose of the use case is to show how to apply the methodology on a simplified but realistic problem, and to provide practical guidance to probabilistic ontology designers on how to apply the UMP-ST. This paper goes beyond previous work on UMP-ST (e.g., *Haberlin, 2013*; *Carvalho et al., 2011*; *Alencar, 2015*) to provide a comprehensive explanation of the methodology in the context of application to a real world problem, along with pragmatic suggestions for how to apply the methodology in practice. For purpose of exposition, our focus is primarily on the procurement fraud use case, but the UMP-ST is applicable to any domain in which semantic technology can be applied.

Our purpose is not to provide rigorous scientific evidence for the value of UMP-ST in comparison with any other methodology or no methodology. *De Hoog (1998)* says, "it is extremely difficult to judge the value of a methodology in an objective way. Experimentation is of course the proper way to do it, but it is hardly feasible because there are too many conditions that cannot be controlled." The value of a methodology like UMP-ST lies in its ability to support a complex system development effort that extends over a long period of time and requires significant resources to implement. Besides the large number of uncontrolled variables, the resources to implement a single case of sufficient complexity is difficult; experimentation requiring multiple parallel implementations is prohibitive. Nevertheless, the experience of our development team is that the structure provided by UMP-ST was essential to the ability to capture the expert's knowledge in a model whose results were a reasonable match to the expert's judgments.

The paper is organized as follows. The next section reviews existing design methodologies that provided inspiration for UMP-ST. The following section introduces the UMP-ST. Next, we introduce our use case devoted to identifying fraud in public procurement in Brazil. The fifth section explains the four disciplines of the UMP-ST in the context of the fraud use case, and is followed by a section discussing applicability of the UMP-ST to other domains. The paper concludes with a section on future work and a final section presenting our concluding remarks.

## RELATED WORK

Successful development of any complex system requires following a structured, systematic process for design, implementation and evaluation. Existing software and systems

engineering processes are useful as starting points, but must be tailored for engineering a probabilistic ontology. The UMP-ST draws upon a number of related processes for software engineering, ontology engineering, and Bayesian network engineering to provide a process tailored to probabilistic ontology engineering. To provide a context for introducing the UMP-ST, this section reviews related literature on design processes that provided an initial basis for tailoring the UMP-ST.

## The unified process

The Unified Process (UP) is a widely applied software engineering process (*Jacobson, Booch & Rumbaugh, 1999*; *Kruchten, 2000*; *Balduino, 2007*). It has three main characteristics: (1) it is iterative and incremental; (2) it is architecture centric; and (3) it is risk focused. Each project is divided into small chunks, called *iterations*, each concluding in delivery of executable code. These frequent deliverables yield an incremental implementation of the system. A key deliverable is the executable architecture, which is a partial implementation of the system that validates the architecture and builds the foundation of the system. Finally, the UP mitigates risk by prioritizing the highest risk features for early implementation. The reasoning is simple: if a critical aspect of the system is going to fail, it is better to discover this early enough to rework the design or cancel the project, than to realize after the fact that large amounts of resources have been wasted on a non-viable project.

The UP defines the *project lifecycle* as composed of four phases: (1) Inception; (2) Elaboration; (3) Construction; and (4) Transition. Inception is usually the shortest phase. The main goal is to define the justification for the project, its scope, the risks involved, and the key requirements. In the elaboration phase, the primary concerns are to define most of the requirements, to address the known risks, and to define and validate the system architecture. The Construction phase is the longest phase, where most of the development process resides. This phase is usually broken down into small iterations with executable code being delivered at the end of each iteration. Finally, in the Transition phase the system is deployed, the users are trained, and initial feedback is collected to improve the system.

To support the project lifecycle, the UP defines several *disciplines* or workflows. Each discipline describes a sequence of activities in which actors or workers produce products or artifacts to achieve a result of observable value. For example, a developer might carry out a programming activity using the system specification in order to produce both source and executable code. There are several variations of the Unified Process (e.g., Rational Unified Process, Agile Unified Process, Enterprise Unified Process). While each has its own set of disciplines, the following disciplines are common to most: Business Modeling, responsible for documenting the business processes in a language common to both business and software communities; Requirements, responsible for defining what the system should do based on the information gathered from the customer; Analysis & Design, responsible for showing how the system will be realized in the implementation phase; Implementation, responsible for developing the code necessary to implement the elicited requirements; Test, which verifies and validates the code developed; and Deployment, responsible for delivering the software to the end user.

## Ontology engineering

According to *Gomez-Perez, Corcho & Fernandez-Lopez (2004)*, the first workshop on Ontological Engineering was held in conjunction with the 12тн European Conference on Artificial Intelligence in 1996. Since then, several methodologies for building ontologies have been proposed.

*Gomez-Perez, Corcho & Fernandez-Lopez (2004)* compare several different methodologies used for building ontologies in the context of the METHONTOLOGY ontology development process (*Fernández-López, Gómez-Pérez & Juristo, 1997*). METHONTOLOGY identifies activities that are performed to build ontologies. The three main activity categories are: (1) ontology management activities; (2) ontology development oriented activities; and (3) ontology support activities.

The *ontology management activities* include: (1) scheduling, which identifies activities, their dependency, the resources needed in each, and how long they will take; (2) control, which guarantees that the project is going according to schedule; and (3) quality assurance, which verifies if the products generated from the scheduled activities are satisfactory. These activities are general enough that they can be imported from other frameworks that are not specific to ontology engineering, such as the Project Management Body of Knowledge (PMBoK), which is a general guide to project management. PMBoK includes activities such as scheduling, control, among others (*Project Management Institute, 2004*). Because these are generic activities, it is not surprising that only one, the On-To-Knowledge (OTKM) methodology (*Sure, Staab & Studer, 2004*), out of seven methodologies analyzed and compared by *Gomez-Perez, Corcho & Fernandez-Lopez (2004)* describes these activities in detail. The METHONTOLOGY methodology only proposes these activities, but does not describe them in detail.

The *ontology development oriented activities* are divided into three different steps: (1) pre-development; (2) development; and (3) post-development activities. Pre-development involves: (1a) an environment study to understand where the ontology will be used, which applications will use it, etc.; and (1b) a feasibility study in order to assess if it is worthwhile, feasible, and cost-effective to build this ontology. Although these are important activities, they are not addressed in most of the methodologies for building ontologies. According to *Gomez-Perez, Corcho & Fernandez-Lopez (2004)* only the METHONTOLOGY methodology proposes the environment study and describes the feasibility study.

Development activities include: (2a) the specification activity, which describes why the ontology is being built, its purpose, etc.; (2b) the conceptualization activity, which describes and organizes the domain knowledge; (2c) the formalization activity, which evolves the conceptual model into a more formal model; and (2d) the implementation activity, which creates the desired ontology in the chosen language. As expected, these are the main activities addressed by the ontology engineering methodologies. The methodologies analyzed by *Gomez-Perez, Corcho & Fernandez-Lopez (2004)* proposed or described most of these development activities, with the exception of Cyc (*Reed & Lenat, 2002*), which only addresses the implementation activity and does not mention the others.

Post-development activities involve (3a) maintenance, which updates and fixes the ontology, and (3b) (re)use of the ontology being developed by other ontologies and

applications. These are also important activities; however, most of the methodologies only address them as a natural step during the ontology's life cycle, which can be incremental, producing a sequence of evolving prototypes. None of the methodologies presented by *Gomez-Perez, Corcho & Fernandez-Lopez (2004)* describes these activities. Only the METHONTOLOGY and OTKM methodologies propose some of these activities, but do not provide much detail.

Finally, *ontology support activities* include: (1) knowledge acquisition, which extracts domain knowledge from subject matter experts (SMEs) or through some automatic process, called ontology learning; (2) evaluation, in order to validate the ontology being created; (3) integration, which is used when other ontologies are used; (4) merging, which is important for creating a new ontology based on a mix of several other ontologies from the same domain; (5) alignment, which involves mapping different concepts to/from the involved ontologies; (6) documentation, which describe all activities completed and products generated for future reference; and (7) configuration management, which controls the different versions generated for all ontologies and documentation. Out of the seven methodologies compared by *Gomez-Perez, Corcho & Fernandez-Lopez (2004)*, five neither propose nor mention configuration management, merging, or alignment. The integration activity is proposed by six of them, but not described in detail. The knowledge acquisition and documentation activities are proposed by three and described by two, while the evaluation activity is proposed and described by two in detail.

## Probability elicitation

The literature on eliciting probabilities from experts has a long history (e.g., *Winkler, 1967*; *Huber, 1974*; *Wallsten & Budescu, 1983*). At the interface between cognitive science and Bayesian probability theory, researchers have examined biases in unaided human judgment (e.g., *Kahneman, Slovic & Tversky, 1982*) and have devised ways to counteract those biases (e.g., *Clemen & Reilly, 2004*; *Burgman et al., 2006*). Several authors have defined structured processes or protocols for eliciting probabilities from experts (e.g., *Clemen & Reilly, 2004*; *Garthwaite, Kadane & O'Hagan, 2005*). There is general agreement on the steps in the elicitation process. The seven steps described by *Clemen & Reilly (2004)* are: understanding the problem; identifying and recruiting experts; motivating the experts; structuring and decomposition; probability and assessment training; probability elicitation and verification; and aggregating the probabilities. A recent comprehensive reference for probability elicitation is *O'Hagan et al. (2006)*.

The advent of graphical probability models (*Pearl, 1988*) has created the problem of eliciting the many probabilities needed to specify a graphical model containing dozens to hundreds of random variables (cf., *Druzdzel & Van der Gaag, 2000*; *Renooij, 2001*). *Mahoney & Laskey (1998)* defined a systematic process for constructing Bayesian network models. Their process considered elicitation of structural assumptions as well as probability distributions. It is an iterative and incremental process that produces a series of prototype models. The lessons learned from building each prototype model are used to identify requirements for refining the model during the next cycle.

# UNCERTAINTY MODELING PROCESS FOR SEMANTIC TECHNOLOGY

The process of creating and using a probabilistic ontology typically occurs in three stages: first is modeling the domain; next is populating the model with situation-specific information; and third is using the model and situation-specific information for reasoning. Modeling a domain means constructing a representation of aspects of the domain for purposes of understanding, explaining, predicting, or simulating those aspects. For our purposes, the model represents the kinds of entities that can exist in the domain, their attributes, the relationships they can have to each other, the processes in which they can participate, and the rules that govern their behavior. It also includes uncertainties about all these aspects. There are many sources of uncertainty: e.g., causes may be non-deterministically related to their effects; events may be only indirectly observable through noisy channels; association of observations to the generating events may be unknown; phenomena in the domain may be subject to statistical fluctuation; the structure of and associations among domain entities may exhibit substantial variation; and/or the future behavior of domain entities may be imperfectly predictable (e.g., *Schum & Starace, 2001*; *Laskey & Laskey, 2008*; *Costa et al., 2012*). Once these and other relevant sources of uncertainty are captured in a domain model, the model can be applied to a specific situation by populating it with data about the situation. Finally, the inference engine can be called upon to answer queries about the specific situation. Unlike traditional semantic systems that can handle only deterministic queries, queries with a probabilistic ontology can return soft results. For example, consider a query about whether an inappropriate relationship exists between a procurement official and a bidder. A reasoning system for a standard ontology can return only procurements in which such a relationship can be proven, while a reasoner for a probabilistic ontology can return a probability that such a relationship exists.

The UMP-ST is an iterative and incremental process, based on the UP, for designing a probabilistic ontology. While UP serves as the starting point, UMP-ST draws upon and is consistent with the ontology engineering and probability elicitation processes described in the previous sections, thus tailoring the UP for probabilistic ontology design.

As shown in Fig. 1, the UMP-ST includes all phases of the UP, but focuses only on the Requirements, Analysis & Design, Implementation, and Test disciplines. The figure depicts the intensity of each discipline during the UMP-ST. Like the UP, UMP-ST is iterative and incremental. The basic idea behind iterative enhancement is to model the domain incrementally, allowing the modeler to take advantage of what is learned during earlier iterations of the model in designing and implementing later iterations. For this reason, each phase includes all four disciplines, but the emphasis shifts from requirements in the earlier phases toward implementation and test in the later phases. Note that testing occurs even during the Inception phase, prior to beginning the implementation phase. This is because it is usually possible to test some aspects of the model during the Analysis & Design

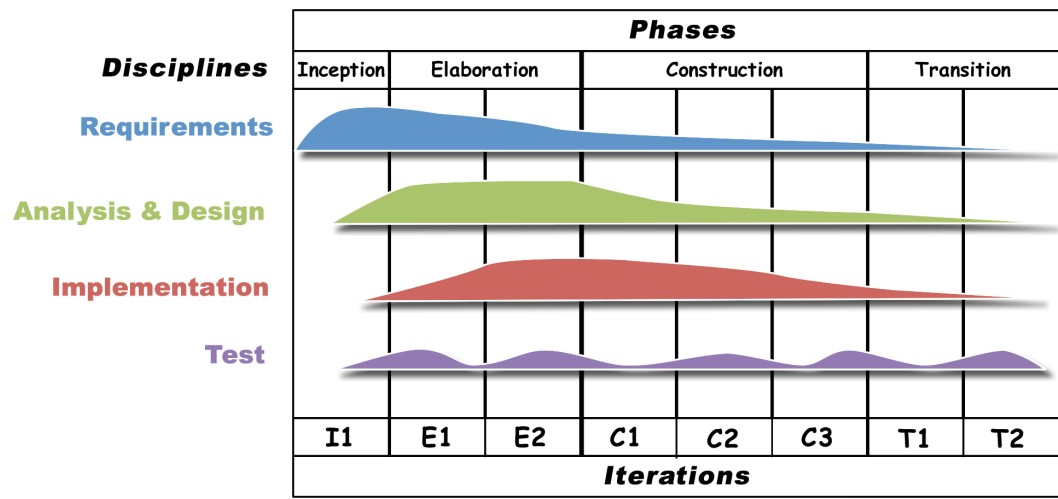

**Figure 1** Uncertainty Modeling Process for Semantic Technology (UMP-ST).

stage prior to implementation. It is well known that early testing reduces risk, saves cost, and leads to better performance (*INCOSE, 2015*).

Figure 2 presents the Probabilistic Ontology Modeling Cycle (POMC). This cycle depicts the major outputs from each discipline and the natural order in which the outputs are produced. Unlike the waterfall model (*Royce, 1970*), the POMC cycles through the steps iteratively, using what is learned in one iteration to improve the result of the next. The arrows reflect the typical progression, but are not intended as hard constraints. Indeed, it is possible to have interactions between any pair of disciplines. For instance, it is not uncommon to discover a problem in the rules defined in the Analysis & Design discipline during the activities in the Test discipline. As a result, the engineer might go directly from Test to Analysis & Design in order to correct the problem.

In Fig. 2, the Requirements discipline (blue box) defines the goals that must be achieved by reasoning with the semantics provided by our model. Usually, when designing a PO, one wants to be able to automate a reasoning process that involves uncertainty. By goals, we mean the kinds of questions the user wants the system to be able to answer via the PO reasoning. For instance, one of the main goals in the procurement fraud domain is to be able to answer with a certain degree of certainty whether a procurement presents any signs of fraud. However, this type of question is not straight-forward to answer. Thus, the system will typically need to evaluate a set of more specific questions, or queries, in order to better assess the probability of having fraud. Furthermore, in order to answer these more specific queries, the system will need some evidence. These goals, queries, and evidence comprise the requirements for the model being designed.

The Analysis & Design discipline (green boxes) describes classes of entities, their attributes, how they relate to each other, and what rules apply to them in our domain. These definitions are independent of the language used to implement the model.

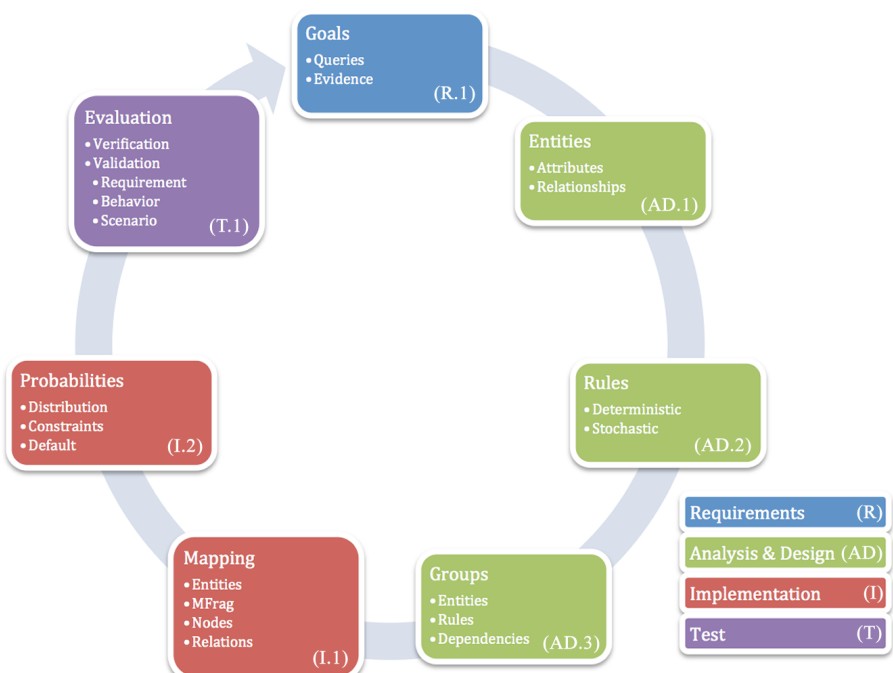

**Figure 2** Probabilistic Ontology Modeling Cycle (POMC)—Requirements in blue, Analysis & Design in green, Implementation in red, and Test in purple.

The Implementation discipline (red boxes) maps the design to a specific language that is both semantically rich and capable of representing uncertainty. This means encoding the classes, attributes, relationships and rules in the chosen language. For our case study, the mapping is to PR-OWL (*Carvalho, Laskey & Costa, 2013*; *Costa, Laskey & Laskey, 2008*), but other semantically rich uncertainty representation languages could also be used (e.g., *Cozman & Mauá, 2015*).

Finally, the Test discipline (purple box) is responsible for evaluating whether the model developed during the Implementation discipline is behaving as expected from the rules defined during Analysis & Design and whether the results achieve the goals elicited during the Requirements discipline. As noted previously, it is a good idea to test some of the rules and assumptions even before implementation. This is a crucial step to mitigate risk. Early testing can identify and correct problems before significant resources have been spent developing a complex model that turns out to be inadequate.

Like several of the ontology engineering processes considered by *Gomez-Perez, Corcho & Fernandez-Lopez (2004)*, the UMP-ST does not cover ontology management, under the assumption that these activities can be imported from other frameworks. Although the UMP-ST does not cover maintenance and reuse, its iterative nature supports incremental evolution of the developed ontology. Of the ontology support activities described by *Gomez-Perez, Corcho & Fernandez-Lopez (2004)*, the UMP-ST process explicitly addresses only the test discipline, which is similar to the evaluation activity. By following the steps in the UMP-ST, the ontology designer will be generating the documentation needed in order to describe not only the final PO, but also the whole process of building it. This

supports the documentation activity of *Gomez-Perez, Corcho & Fernandez-Lopez (2004)*. Like most ontology engineering processes, the UMP-ST does not address the ontology support activities of integration, merging, and alignment.

The primary focus of the UMP-ST is the ontology development activities. Because it is based on the UP, it uses a different nomenclature than *Gomez-Perez, Corcho & Fernandez-Lopez (2004)*, but there is a close resemblance: the specification activity is similar to the requirements discipline; the conceptualization and formalization activities are similar to the analysis & design discipline; and the implementation activity is similar to the implementation discipline. The major difference between the methodologies reviewed by *Gomez-Perez, Corcho & Fernandez-Lopez (2004)* and the UMP-ST is the focus. While *Gomez-Perez, Corcho & Fernandez-Lopez (2004)* focus on ways to build a glossary of terms, build taxonomies, and define concepts, properties, and deterministic rules the UMP-ST presents techniques to identify and specify probabilistic rules, define dependency relations between properties based on these rules, and quantify the strength of these relations as parameters of local probability distributions. Thus, the UMP-ST extends other methodologies used for building ontologies, and should coexist with these methodologies. When creating deterministic parts of the ontology the user can follow existing methodologies proposed for standard ontology building. To incorporate uncertainty and therefore extend to a probabilistic ontology, the user can follow the steps defined in the UMP-ST process.

Similarly, the UMP-ST can and should coexist with processes for eliciting probabilities, such as those defined by *Clemen & Reilly (2004)* and *O'Hagan et al. (2006)*. The probabilistic ontology engineer should refer to these resources when defining a protocol for eliciting probabilities from experts.

In the next two sections, the UMP-ST process and the POMC are illustrated through a case study in procurement fraud detection and prevention. The case study walks step-by-step through the activities that must be executed in each discipline in the POMC. The case study has been kept simple enough for clear exposition of POMC, while being complex enough to convey key issues that arise in real-world ontology engineering. Implementation and plausible reasoning were carried out using the UnBBayes probabilistic ontology environment (*Carvalho, 2008*; *Matsumoto et al., 2011*).

## PREVENTING AND DETECTING PROCUREMENT FRAUD IN BRAZIL

In Brazil, the main law that details the regulation of the public procurement process is the Federal Law 8,666/93 (Public Procurement Law). The public procurement procedures are also mentioned in Section XXI, Article 37 of the Federal Constitution. The Public Procurement Law is applicable not only to the Federal Government, but also to the State and Municipal Governments. Although it is meant to provide just general guidelines, it is so detailed that there is little room for the States and Municipalities to further legislate (*Frizzo & Oliveira, 2014*). The Public Procurement Law regulates public procurement procedures and contracts involving the government.

The Public Procurement Law defines three main procurement procedures: invitation to tender, which is a simpler and faster procedure where at least three competitors are invited to participate in the tender based on the request for proposals (RFP), which is not required to be advertised in the press; price survey, which requires the competitors to be previously registered before the public tender and requires a broader advertising of the RFP in the newspaper and official press; and competition, which is the most complex and longest procedure, allowing participation of all companies that meet the qualification criteria on the first day of the procedure and requiring more general advertising of the RFP as the price survey. In addition, Law 10,520/02 created the reverse auction, which involves alternative bids from the participating companies in the competitive phase, before the qualification documents are analyzed. Nowadays, the most common procedure for the acquisition of common goods and services is the electronic reverse auction, which is the same as the reverse auction, but the procedure happens in an electronic system. Its RFP must also be advertised in the official press as well as through the Internet.

The criteria for selecting the best proposal are defined in the RFP by the regulated agency. There are three main types of rules that must be followed: best price, where the company that presents the best bid and meets the minimum requirements is awarded the contract; best technique, where the company with the best technical solutions wins regardless of price; and a mix of the two, where scores are given for both price and technique and the company with the highest joint score wins.

*Frizzo & Oliveira (2014)* provide additional detail on thresholds for determining whether a contract is subject to the Public Procurement Law, freedom to choose which procedure to use, changes to an existing contract, and other aspects of the public procurement process in Brazil.

The procurement process presents many opportunities for corruption. Although laws attempt to ensure a competitive and fair process, perpetrators find ways to turn the process to their advantage while appearing to be legitimate. To aid in detecting and deterring such perversions of the procurement process, a specialist, who helped in this work, has didactically structured different kinds of procurement fraud encountered by the Brazilian Office of the Comptroller General (CGU, *Controladoria-Geral da União*, in Portuguese) over the years.

These different fraud types are characterized by criteria, such as business owners working as a front for the company or use of accounting indices that are not commonly employed. Indicators have been established to help identify cases of each of these fraud types. For instance, one principle that must be followed in public procurement is that of competition. A public procurement should attempt to ensure broad participation in the bidding process by limiting requirements on bidders to what is necessary to guarantee adequate execution of the contract. Nevertheless, it is common to have a fake competition in which different bidders are, in fact, owned by the same person. This is usually done by having someone act as a front for the enterprise. An indicator that a bidder may be a front is that the listed owner has little or no education. Thus, an uneducated owner is a red flag suggesting that there may be a problem with the procurement.

*Gregorini (2009)* identified a number of red flags that can be considered evidence of fraud. These include: concentration of power in the hands of a few people; rapid growth in concentration of goods and services contracted from a single company; competition restriction; transfer of funds to a Non Governmental Organization (NGO) close to elections; and others. While these factors are evidence of potential irregularities, they are not definitive indicators. A list of more serious and determinant conditions is presented by *Flores (2004)*. These include: choosing directors based on a political agenda; negotiating contracts in order to reserve money for an election campaign; negotiating contracts in order to favor friends and family; bribery in order to obtain certain privileges; and providing inside information.

A more formal definition of different types of fraud found in Brazil is presented by *Oliveira (2009)*. He presents three main groups of fraud, based on recent scandals in Brazil: frauds initiated by passive agents; frauds initiated by active agents; and frauds that represent collusion. The first is when an agent from the Public Administration, acting in his public function, favors someone or himself by performing illicit actions (e.g., purchasing products that were never used, falsification of documents and signatures, favoring friends and family). The second is when an active agent, a person or a company, outside the Public Administration tries to corrupt an agent that works in the Public Administration or does something illegal in order to cheat the procurement process (e.g., acting as a front for a company, delivering contraband products, giving money to civil servants in order to favor a specific company). Finally, the third is when there is some type of collusion between companies participating in the procurement process or even between passive and active agents (e.g., delivering and accepting just part of the goods purchased, paying before receiving the merchandise, overpricing goods and services, directing and favoring a specific company in exchange for some financial compensation).

The types of fraud presented by *Oliveira (2009)*, although focused on the Brazilian context, are consistent with more recent work from *Dhurandhar et al. (2015b)*. This work, which presents a more general fraud taxonomy related to procurement fraud, was registered as a patent in 2015 (*Dhurandhar et al., 2015a*). While Oliveira talks about passive and active agents, Dhurandhar et al. talks about fraud by employees and fraud by vendors, respectively. However, these fraud definitions do have a few differences. For example, while Dhurandhar et al. differentiate collusion among vendors and collusion between employee and vendors, Oliveira classifies both as simply collusion.

Formalizing knowledge about fraud in a computable form can lead to automated support for fraud detection and prevention. Specifically, analysts at the CGU must sift through vast amounts of information related to a large number of procurements. Automated support can improve analyst productivity by highlighting the most important cases and the most relevant supporting information. The ultimate goal of the procurement fraud probabilistic ontology is to structure the specialist's knowledge to enable automated reasoning from indicators to potential fraud types. Such an automated system is intended to support specialists and to help train new specialists, but not to replace them. Automated support for this task requires a semantically rich representation that supports uncertainty management.

As a case study, *Carvalho (2011)* developed a proof-of-concept probabilistic ontology covering part of the procurement fraud domain. This paper uses a portion of this case study to illustrate how the POMC can support the creation of a PO. The full implementation and code for the case study is presented in *Carvalho* (*2011*) and is provided as Supplemental Information. This proof-of-concept implementation represents only a fragment of the specialist's knowledge of the procurement fraud domain. The plan is eventually to extend this PO to a full representation of the specialist's knowledge.

## UMP-ST FOR PROCUREMENT FRAUD

This section describes in detail the four disciplines in the UMP-ST process and their application to the procurement fraud case study. To facilitate the understanding of each discipline, we alternate between describing the discipline and illustrating its application to the public procurement fraud detection and prevention use case.

### Requirements

The POMC begins with the Requirements discipline (R.1 in Fig. 2). The requirements discipline defines the objectives that must be achieved by representing and reasoning with a computable representation of domain semantics. For this discipline, it is important to define the questions that the model is expected to answer, i.e., the queries to be posed to the system being designed. For each question, a set of information items that might help answer the question (evidence) must be defined.

Requirements can be categorized as functional and non-functional (*Wiegers, 2003*; *Sommerville, 2010*). Functional requirements concern outputs the system should provide, features it should have, how it should behave, etc. In our case, functional requirements relate to the goals, queries, and evidence that pertain to our domain of reasoning. Non-functional requirements, on the other hand, address criteria relating to performance of the system as a whole. For instance, in our use case a non-functional requirement could be that a given query has to be answered in less than a minute. Another example is that the posterior probability given as an answer to a given query has to be either exact or an approximation with an error bound of $\pm 0.5\%$. Non-functional requirements are typically not specific to probabilistic ontology development. We therefore focus here on how to develop functional requirements for our use case.

We focus on a subset of our procurement use case to illustrate how a requirement is carried through the PO development cycle until it is eventually implemented and tested. To understand the requirements associated with this subset, we first have to explain some of the problems encountered when dealing with public procurements.

One of the principles established by Law No. 8,666/93 is equality among bidders. This principle prohibits the procurement agent from discriminating among potential suppliers. However, if the procurement agent is related to the bidder, he/she might feed information or define new requirements for the procurement in a way that favors the bidder.

Another problem arises because public procurement is quite complex and may involve large sums of money. Therefore, members forming the committee for a procurement must both be well prepared, and have a clean history with no criminal or administrative

convictions. This latter requirement is necessary to satisfy the ethical guidelines that federal, state, municipal and district government employees must follow.

The above considerations give rise to the following set of goals, queries, and evidence:

1. *Goal*: identify whether a given procurement violates fair competition policy (i.e., evidence suggests further investigation and/or auditing is warranted);
   (a) *Query*: is there any relation between the committee and the enterprises that participated in the procurement?
      i. *Evidence*: committee member and responsible person of an enterprise are related (mother, father, brother, or sister);
      ii. *Evidence*: committee member and responsible person of an enterprise live at the same address.
2. *Goal*: identify whether the committee for a given procurement has improper composition.
   (a) *Query*: is there any member of committee who does not have a clean history?
      i. *Evidence*: committee member has criminal history;
      ii. *Evidence*: committee member has been subject to administrative investigation.
   (b) *Query*: is there any relation between members of the committee and the enterprises that participated in previous procurements?
      i. *Evidence*: member and responsible person of an enterprise are relatives (mother, father, brother, or sister);
      ii. *Evidence*: member and responsible person of an enterprise live at the same address.

In defining requirements, the availability of evidence must be considered. For example, information about whether persons are related might be drawn from a social network database; evidence about criminal history might come from a police database; an evidence about cohabitation might be drawn from an address database. One important role for semantic technology is to support interoperability among these various data sources and the fraud detection model.

Another important aspect of the Requirements discipline is defining traceability of requirements. According to *Gotel & Finkelstein (1994)*, "requirements traceability refers to the ability to describe and follow the life of a requirement, in both the forward and backward directions." A common tool for traceability is a specification tree, in which each requirement is linked to its "parent" requirement. A specification tree for the requirements for our procurement model is shown in Fig. 3. In this hierarchy, each item of evidence is linked to a query it supports, which in turn is linked to its higher level goal. This linkage supports requirements traceability.

In addition to the hierarchical decomposition of the specification tree, requirements should also be linked to work products of other disciplines, such as the rules in the Analysis & Design discipline, or goals, queries, and evidence elicited in the Requirements discipline. These links provide traceability that is essential to validation and management of change. Subsequent sections show how UMP-ST supports requirements tracing.

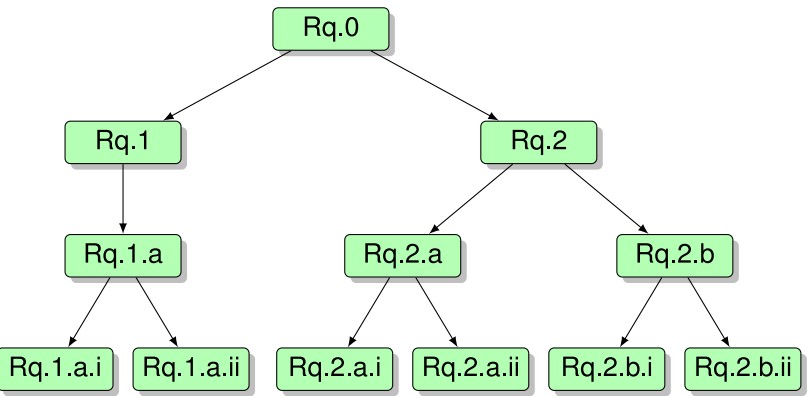

**Figure 3** Specification tree for procurement model requirements.

## Analysis & Design

Once we have defined our goals and described how to achieve them, it is time to start modeling the entities, their attributes, relationships, and rules to make that happen. This is the purpose of the Analysis & Design discipline.

The major objective of this discipline is to define the semantics of the model. In fact, much of the semantics can be defined using traditional ontologies, including the deterministic rules that the concepts described in our model must obey. The focus of this paper is on representing uncertain aspects of the domain. Information on defining traditional ontologies can be found in *Allemang & Hendler (2011)* and *Gomez-Perez, Corcho & Fernandez-Lopez (2004)*.

The first step in defining the domain model is to define the classes and relationships that are important to represent for the procurement fraud detection problem (AD.1 in Fig. 2). For our case study, we use the Unified Modeling Language (UML) (*Rumbaugh, Jacobson & Booch, 1999*) for this purpose. Analysis & Design also includes developing rules (AD.2 in Fig. 2). Because UML is insufficiently expressive to represent complex rule definitions, we record the deterministic rules separately for later incorporation into the PR-OWL probabilistic ontology. While experienced ontology engineers might prefer to define classes, relationships and rules directly in OWL, we chose UML for its popularity, understandability, ease of communication with domain experts, and widely available and usable software tools. We see UML-style diagrams as a way to capture knowledge about classes and relationships that could be automatically translated into an OWL ontology or PR-OWL probabilistic ontology (*cf.*, *Gasevic et al. (2004)*).

Figure 4 depicts a simplified model of the classes and relationships in the procurement fraud domain. A Person has a name, a mother and a father (also Person). Every Person has a unique identification that in Brazil is called CPF. A Person also has an Education and livesAt a certain Address. In addition, everyone is obliged to file his/her TaxInfo every year, including his/her annualIncome.[1] These entities can be grouped as Personal Information. A PublicServant is a Person who worksFor a PublicAgency, which is a Government Agency. Every public Procurement is owed by a PublicAgency, has a committee formed by a group of PublicServants, and has a

---

[1] Every Brazilian citizen is required to file tax information, even if only to state that his or her income is below a certain amount and no taxes are owed.

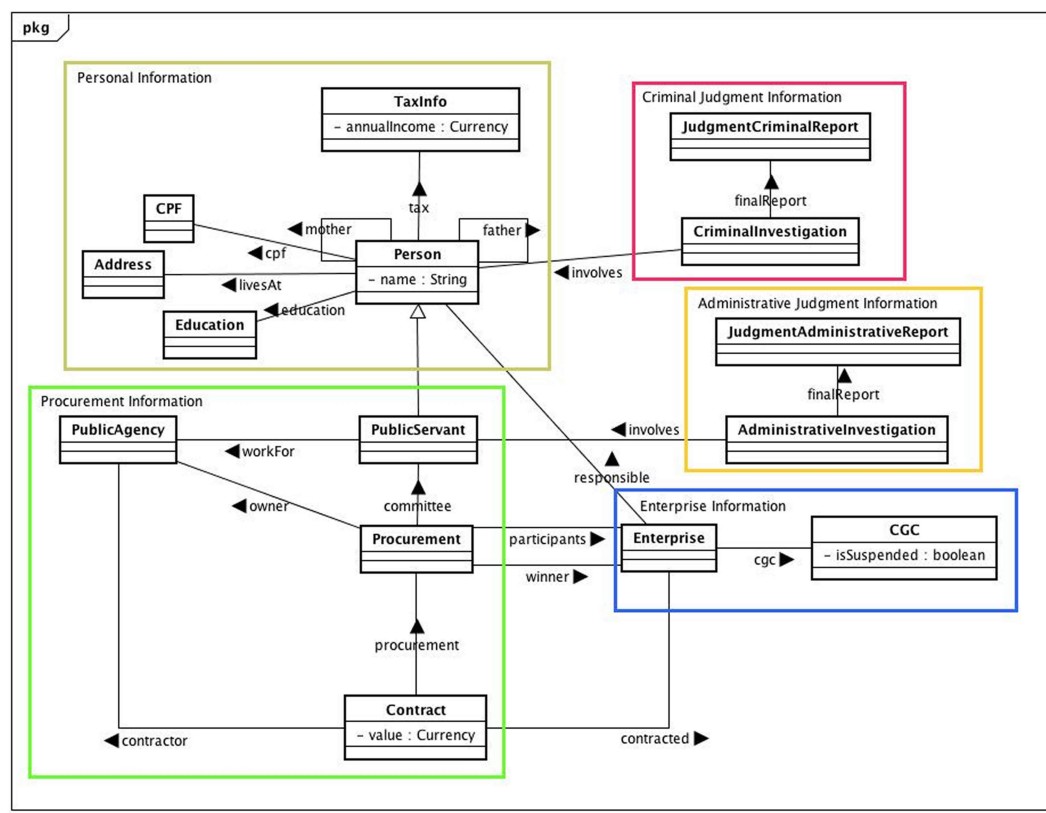

**Figure 4  Entities, their attributes, and relations for the procurement model.**

group of `participants`, which are `Enterprises`. One of these will be the `winner` of the `Procurement`. Eventually, the `winner` of the `Procurement` will receive a `Contract` of some `value` with the `PublicAgency` owner of the `Procurement`. The entities just described can be grouped as `Procurement Information`. Every `Enterprise` has at least one `Person` that is `responsible` for its legal acts.

An `Enterprise` also has an identification number, the General List of Contributors `CGC`, which can be used to inform that this `Enterprise` is suspended from procuring with the public administration, `isSuspended`. These are grouped as the `Enterprise Information`. We also have `AdminstrativeInvestigation`, which has information about investigations that involve one or more `PublicServer`. Its `finalReport`, the `JudgmentAdministrativeReport`, contains information about the penalty applied, if any. These entities form the `Administrative Judgment Information`. Finally we have the `Criminal Judgment Information` group that describes the `CriminalInvestigation` that involves a `Person`, with its `finalReport`, the `JudgmentCriminalReport`, which has information about the verdict.

Notice that just a subset of this UML model is of interest to us in this paper, since we are dealing with just a subset of the requirements presented in *Carvalho (2011)*.

In addition to the cardinality and uniqueness rules defined above for the entities depicted in Fig. 4, the AD.2 step in Fig. 2 includes specifying probabilistic rules to address the requirements defined in the R.1 step. These include:

1. If a member of the committee has a relative (mother, father, brother, or sister) responsible for a bidder in the procurement, then it is more likely that a relation exists between the committee and the enterprises, which inhibits competition.
2. If a member of the committee lives at the same address as a person responsible for a bidder in the procurement, then it is more likely that a relation exists between the committee and the enterprises, which lowers competition.
3. If 1 or 2, then the procurement is more likely to violate policy for fair competition.
4. If a member of the committee has been convicted of a crime or has been penalized administratively, then he/she does not have a clean history. If he/she was recently investigated, then it is likely that he/she does not have a clean history.
5. If the relation defined in 1 and 2 is found in previous procurements, then it is more likely that there will be a relation between this committee and future bidders.
6. If 4 or 5, then it is more likely that the committee violates policy for proper committee composition.

Typically the probabilistic rules are described initially using qualitative likelihood statements. Implementing a probabilistic ontology requires specifying numerical probabilities. Probability values can be elicited from domain experts (e.g., *Druzdzel & Van der Gaag, 2000*; *O'Hagan et al., 2006*) or learned from observation. The growing literature in statistical relational learning (e.g., *Getoor & Taskar, 2007*) provides a wealth of methods for learning semantically rich probability models from observations. In the Analysis & Design stage, information is identified for specifying the probability distributions (expert judgment and/or data sources). This information is encoded into the target representation during the Implementation stage.

The traceability matrix of Table 1 depicts how the probabilistic rules defined above are traced to the goals, queries and evidence items defined in the Requirements discipline. This traceability matrix is an important tool to help designers to ensure that all requirements have been covered. It also supports maintainability by helping ontology engineers to identify how requirements are affected by changes in the model. It is also important at this stage to trace each of the rules to the source of information used to define the rule (e.g., notes from interview with expert, training manual, policy document, data source).

Another important step in the Analysis & Design discipline is to form natural groups of entities, rules, and dependencies (AD.3 in Fig. 2). This step facilitates the Implementation discipline. The more complex the domain, the more important is the grouping activity. As shown in Fig. 4, even in this simplified example there are five natural groups: (1) Personal Information; (2) Procurement Information; (3) Enterprise Information; (4) Administrative Judgment Information; and (5) Criminal Judgment Information.

| Table 1 | Traceability Matrix Relating Rules to Requirements. | | | | | |
|---|---|---|---|---|---|---|
|  | **Rule.1** | **Rule.2** | **Rule.3** | **Rule.4** | **Rule.5** | **Rule.6** |
| Rq.1 | X | X | X |  |  |  |
| Rq.1.a | X | X |  |  |  |  |
| Rq.1.a.i | X |  |  |  |  |  |
| Rq.1.a.ii |  | X |  |  |  |  |
| Rq.2 |  |  |  | X | X | X |
| Rq.2.a |  |  |  | X |  |  |
| Rq.2.a.i |  |  |  | X |  |  |
| Rq.2.a.ii |  |  |  | X |  |  |
| Rq.2.b |  |  |  |  | X |  |
| Rq.2.b.i |  |  |  |  | X |  |
| Rq.2.b.ii |  |  |  |  | X |  |

## Implementation

Once the Analysis & Design step has been completed, the next step is to implement the model in a specific language. How this discipline is carried out depends on the specific language being used. Our case study was developed using the PR-OWL probabilistic ontology language (*Costa, 2005*; *Carvalho, 2008*). PR-OWL (pronounced "prowl") adds new definitions to OWL to allow the modeler to incorporate probabilistic knowledge into an OWL ontology. This section shows how to use PR-OWL to express uncertainty about the procurement fraud domain.

PR-OWL uses Multi-Entity Bayesian Networks (MEBN) (*Laskey, 2008*) to express uncertainty about properties and/or relations defined on OWL classes. A probability model is defined as a set of MEBN Fragments (MFrags), where each MFrag expresses uncertainty about a small number of attributes of and/or relationships among entities. A set of properly defined MFrags taken together comprise a MEBN theory (MTheory), which can express a joint probability distribution over complex situations involving many entities in the domain. Unlike most expressive probabilistic languages that assume the domain is finite (e.g., *Heckerman, Meek & Koller, 2004*), an MTheory can express knowledge about an unbounded or even infinite set of entities. A properly defined PR-OWL model expresses an MTheory, and thus expresses a global joint distribution over the random variables mentioned in the theory. For more detailed explanations on the key features of MEBN logic, the reader should refer to *Laskey (2008)*.

On a typical usage of a PR-OWL probabilistic ontology, during execution time (e.g., in response to a query) a logical reasoning process would instantiate the MFrags that are needed to respond to the query. The result of this process is a situation-specific Bayesian network (SSBN), which is a minimal Bayesian network sufficient to obtain the posterior distribution for a set of target random variable instances given a set of finding random variable instances. In a PR-OWL probabilistic ontology, the entity types correspond to OWL classes, the attributes correspond to OWL properties, and the relationships correspond to OWL relations. Thus, PR-OWL allows the ontology designer to specify probability distributions to express uncertainty about properties and relations in an OWL ontology.

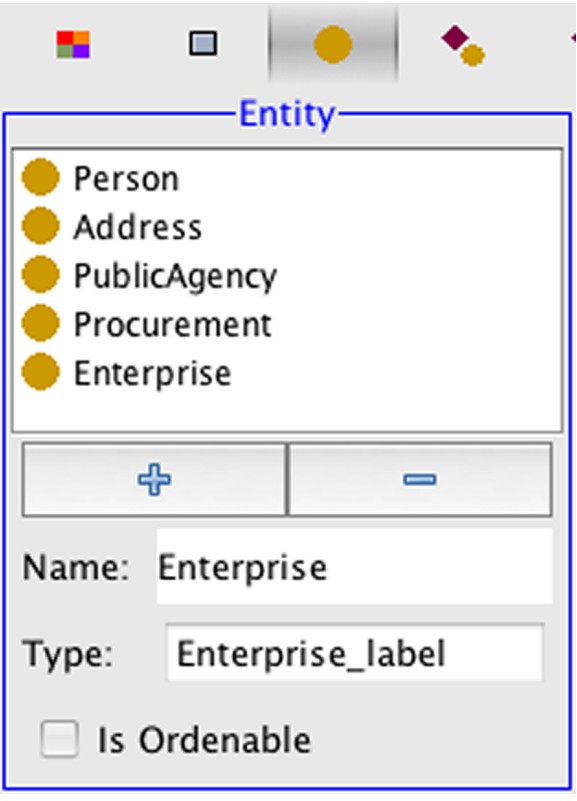

**Figure 5** PR-OWL entities for the procurement domain.

The expressive power of MEBN/PR-OWL makes it an attractive choice for implementing probabilistic ontologies in complex domains. Its compatibility with OWL, a widely used ontology language, allows for the expression of uncertainty in existing OWL ontologies, and for integrating PR-OWL probabilistic ontologies with other ontologies expressed in OWL. These are the primary reasons for the choice of MEBN/PR-OWL as the implementation language in our case study.

The first step in defining a PR-OWL probabilistic ontology for the procurement fraud domain is to represent the entities, attributes and relations of Fig. 4 as OWL classes, properties and relations (I.1 in Fig. 2). Our proof-of-concept made a few simplifications to the representation depicted in Fig. 4. For example, we removed the `PublicServant` entity and connected `Person` directly to `PublicAgency` with the `workFor` relationship. As another simplification, we assumed that every `Person` and `Enterprise` instance is uniquely identified by its name, so there was no need to represent the `CPF` and `CGC` entities. Fig. 5 presents the entities as entered into our PR-OWL ontology implemented in UnBBayes (*Carvalho et al., 2009*).

After defining the entities, we consider characteristics that may be uncertain. An uncertain attribute of an entity or an uncertain relationship among entities is represented in MEBN by a random variable (RV). For example, the RV `livesAt(person)` corresponds

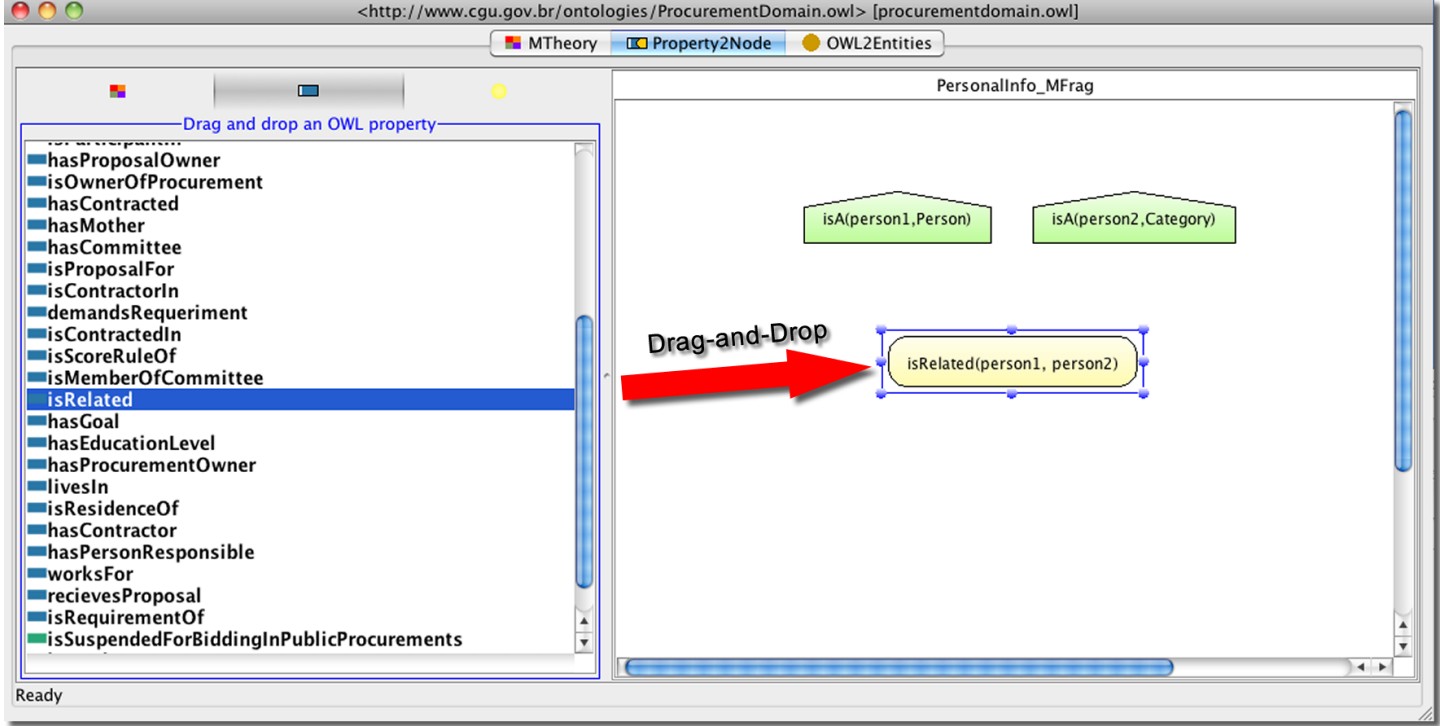

**Figure 6**  Creating a RV in PR-OWL plug-in from its OWL property by drag-and-drop.

to the relation `livesAt` from Fig. 4. As it is a functional relation, `livesAt` relates a `Person` to an `Address`. Hence, the possible values (or states) of this RV are instances of `Address`.

To define a probability distribution for an uncertain attribute or relationship, we must declare it as resident in some MFrag. This occurs as part of I.1 in Fig. 2; its probability distribution will be defined later as part of I.2. For example, Fig. 6 shows how to define uncertainty about whether two persons are related. This is accomplished by selecting the OWL property `isRelated` and dragging the property and dropping it inside the `PersonalInfo` MFrag. The MFrags are language specific groupings formed out of the grouping performed during the Analysis & Design discipline (AD.3 from Fig. 2). The yellow oval on the right-hand side of Fig. 6 shows the RV defined by the PR-OWL plug-in for UnBBayes (*Matsumoto, 2011*) to represent uncertainty about whether persons are related. In the background what actually happens is that an instance of the `DomainResidentNode` class, which is a random variable that has its probability distribution defined in the current MFrag, is created. In addition, an assertion is also added saying that this instance `definesUncertaintyOf` the OWL property `isRelated`.

Once RVs have been created for all uncertain attributes and relationships, probabilistic dependencies can be identified by analyzing how the RVs influence each other. The rules defined as part of the Analysis & Design discipline describe probabilistic relationships that are formally defined as part of the Implementation discipline. For example, rule 2 indicates that there is a dependence between `hasCriminalHistory(person)`, `hasAdministrativeHistory(person)`, and `hasCleanHistory(person)`.

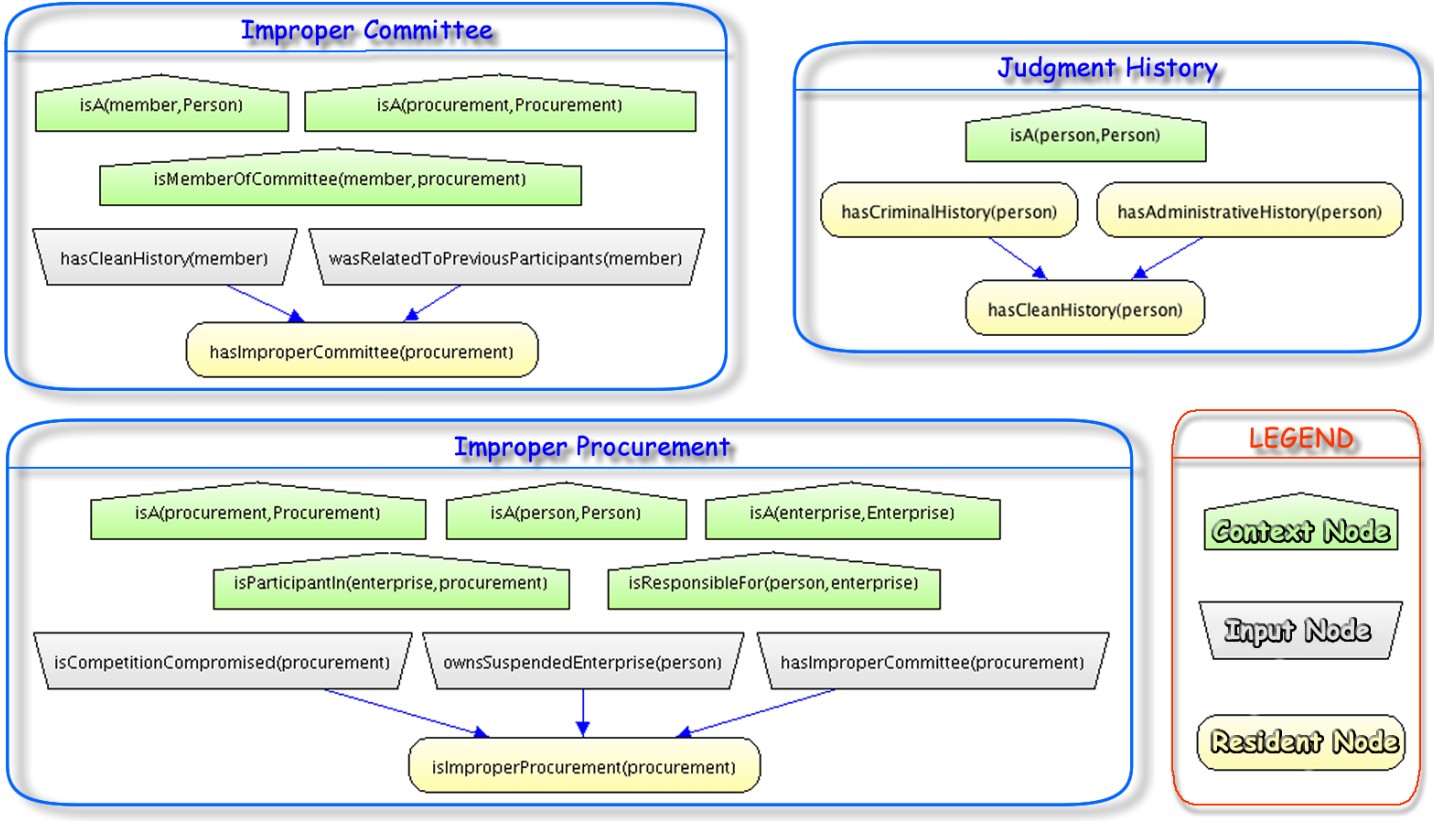

**Figure 7** Part of the probabilistic ontology for fraud detection and prevention in public procurements.

[2] A more sophisticated model for deciding whether to do further investigation or change the committee would define a utility function and use expected utility to make the decision. Future versions of UnBBayes will support Multi-Entity Influence Diagrams (*Costa, 2005*) for modeling decision-making under uncertainty.

[3] Maybe a better name for this node would be `isTrustworthy`. Nevertheless, the idea is that if someone was investigated and/or convicted then he might not be a good candidate for being part of a procurement committee.

For this paper, we focus on the `Judgment History`, `Improper Committee`, and `Improper Procurement` MFrags. Figure 7 shows a partial MTheory consisting of these three MFrags. Details on the complete PR-OWL MTheory can be found in *Carvalho (2011)* and are provided as Supplemental Information.

Each MFrag defines local probability distributions (LPDs) for its *resident* RVs, shown as yellow ovals. These distributions are conditioned on satisfaction of constraints expressed by *context* RVs, shown as green pentagons. The local distributions may depend on the values of *input* RVs, shown as gray trapezoids, whose distributions are defined in the MFrags in which they are resident.

The two main goals described in our requirements are defined in the `Improper Procurement` and `Improper Committee` MFrags.[2] The `Judgment History` MFrag has RVs representing the judgment (criminal and administrative) history of a `Person`.

There are three LPDs defined in the `Judgment History` MFrag: (1) a probability that a person has a criminal history; (2) a probability that a person has an administrative history, and (3) a probability that a person has a clean history given whether or not that person has a criminal and/or an administrative history. This latter probability is lowest if he/she has never been investigated, higher if he/she has been investigated, and extremely high if he/she has been convicted.[3]

The `Improper Committee` MFrag contains the resident RV `hasImproperCommittee` (procurement), defined under the context constraints that `procurement` is an entity of type `Procurement`, `member` is an entity of type `Person`, and `member` is a member of the committee for `procurement`. The assumptions behind the LPD defined in this MFrag are that: if any committee member of this procurement does not have a clean history, or if any committee member was related to previous participants, then the committee is more likely to be improper; and that if these things happen together, the probability of a improper committee is even higher.

The `Improper Procurement` MFrag has the resident RV `isImproperProcurement` (procurement), created in the same way as the `isRelated` RV inside the `PersonalInfo` MFrag explained previously. The assumptions behind the LPD defined in this MFrag are that: if the competition is compromised, or if any owner of a participating enterprise owns a suspended enterprise, or if committee of this procurement is improper, then the procurement is more likely to be improper; and that if these things happen together, the probability of having an improper procurement is even higher.

The final step in constructing a probabilistic ontology in UnBBayes is to define the LPDs for all resident RVs (I.2 in Fig. 2). Figure 8 shows the LPD for the resident node `isImproperProcurement(procurement)`, which is the main question we need to answer in order to achieve one of the main goals in our model. This distribution follows the UnBBayes-MEBN grammar for defining LPDs (*Carvalho, 2008*; *Carvalho et al., 2008*). The distribution for `isImproperProcurement` depends on the values of the parent RVs `isCompetitionCompromised`, `hasImproperCommittee` and `ownsSuspendedEnterprise`. The LPD is defined through a series of if-then-else statements giving the probability of `ownsSuspendedEnterprise` given each combination of truth-values of its parents. In this example, if all three parent RVs are true, then `ownsSuspendedEnterprise` has probability 0.9; if any two parents are true, then `ownsSuspendedEnterprise` has probability 0.8; if just one parent is true, then `ownsSuspendedEnterprise` has probability 0.7; if none of the parents is true then `ownsSuspendedEnterprise` has probability 0.0001. The probability values shown here were defined in collaboration with the specialist who supported the case study. In general, probability values for the MFrags are defined through some combination of expert elicitation and learning from data. However, in the PO described in this paper, all LPDs were defined based on the experience of the SMEs from CGU, since there is not enough structured data to learn the distributions automatically.

It is important to ensure traceability between the MFrags defined in the Implementation stage and the rules defined in the Analysis & Design stage. A traceability matrix similar to Table 1 was developed to trace MFrags to rules. This mapping, along with the mapping of the rules to the requirements as documented in Table 1, enables the probabilistic relationships expressed in the MFrags to be traced back to the requirements defined in the Goals stage.

## Test

As with any engineering methodology, test (T.1 in Fig. 2) plays an essential role in UMP-ST. As *Laskey & Mahoney (2000)* point out, test should do more than showcase the model and

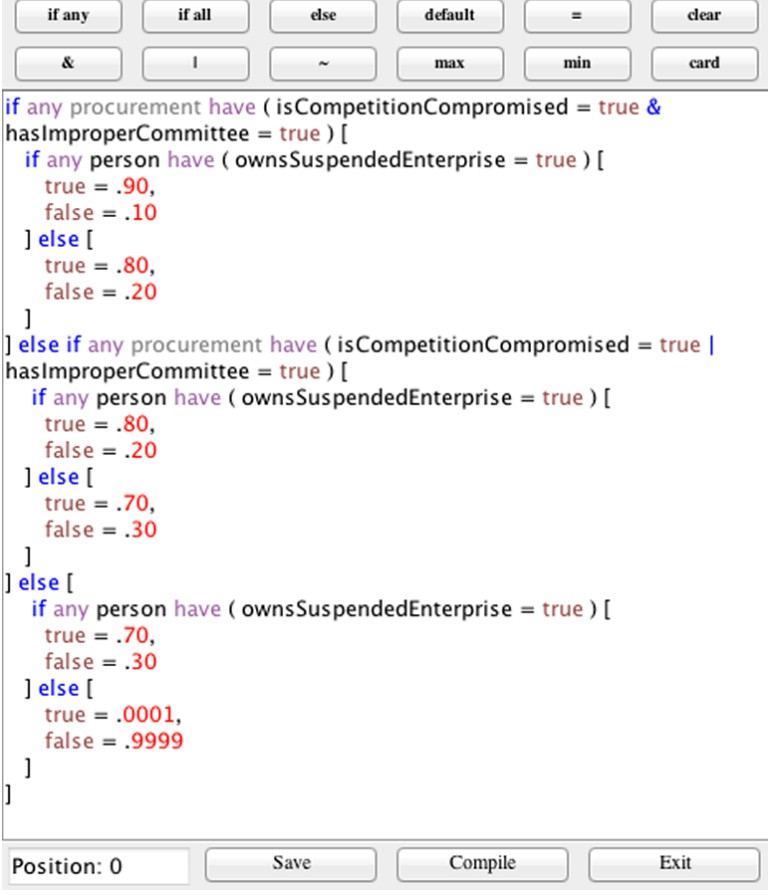

**Figure 8   LPD for node `isImproperProcurement(procurement)`.**

demonstrate that it works as envisioned. Another important goal of the Test discipline is to find flaws and areas for improvement in the model.

The literature distinguishes two types of evaluation, verification and validation (*Adelman, 1992*). Verification is concerned with establishing that "the system was built right," i.e., that the system elements conform to their defined performance specifications. Validation is concerned with establishing that the "right system was built," i.e., that it achieves its intended use in its operational environment.

For example, in the model we have been describing in this Section we would like to verify that the system satisfies the non-functional requirements developed during the Requirements stage as described above, e.g., that the queries covered by the requirement are answered in less than a minute and that the posterior probability given as an answer to a given query is either exact or has an approximation with an error bound of .5% or less.

*Laskey & Mahoney (2000)* present three types of evaluation: elicitation review, importance analysis, and case-based evaluation.

Elicitation review includes reviewing the model documentation, analyzing whether all the requirements were addressed in the final model, making sure all the rules defined during the Analysis & Design stage were implemented, validating the semantics of the concepts

described by the model, etc. This is an important step towards achieving consistency in our model, especially if it was designed by more than one expert. Elicitation review can also confirm that the rules as defined correctly reflect stakeholder requirements.

The traceability matrices are a useful tool for verifying whether all the requirements were addressed in the final implementation of the model. By looking at the matrix tracing MFrags to rules, we can verify that all the rules defined during Analysis & Design have been covered. The traceability matrix of Table 1, defined during Analysis & Design, ensured that the rules covered all the defined requirements. Therefore, by composing these matrices, we can infer that all the requirements have been implemented in our model. This review should also confirm that important stakeholder requirements were not missed during Analysis & Design.

Of course, an initial implementation will often intentionally cover only a subset of the stakeholder requirements, with additional requirements being postponed for later versions. Lessons learned during implementation are reviewed at this stage and priorities for future iterations are revisited and revised.

Importance analysis is a model validation technique described by *Laskey & Mahoney (2000)*. A form of sensitivity analysis, its purpose is to verify that selected parts of the model behave as intended. In importance analysis, one or more *focus* RVs are specified and their behavior is examined under different combinations of values for evidence RVs. The output is a plot for each focus RV that orders the evidence RVs by how much changes in the value of the evidence RV affect the probability of the focus RV. Importance analysis is an important type of unit testing. In the case of PR-OWL, we can analyze the behavior of the random variables of interest given evidence per MFrag. This MFrag testing is important to capture local consistency of the model and to help localize the source of any problems identified in the model.

The tests designed in this section as well as the model described in this paper were developed with the help of experts from the Department of Research and Strategic Information from CGU. They provided detailed information on the different types of frauds as well as on evidence that they usually search for when auditing contracts during the internal control activities. Furthermore, they have also validated the proof-of-concept model described in this paper with the tests we will describe as well as others that were omitted due to space restrictions.

As an example of unit testing, we demonstrate how to define different scenarios to test the `Judgment History` MFrag. Essentially, we want to verify how the query `hasCleanHistory(person)` will behave in light of different set of evidence for a person's criminal and administrative history.

Results for just one combination of states for the parent RVs are shown in Fig. 9, which shows three distinct scenarios for a 3-node model. The model assesses whether or not a given person (*person 1* in the figure) has a clean history. It consists of a binary RV with two parents. Each parent represents whether or not the person has been convicted, investigated, or not investigated in a criminal process (hasCriminalHistory__person1) or in an administrative process (hasAdministrativeHistory__person1). Figure 9A shows the model with no evidence entered (all nodes in yellow with non-zero probabilities), which

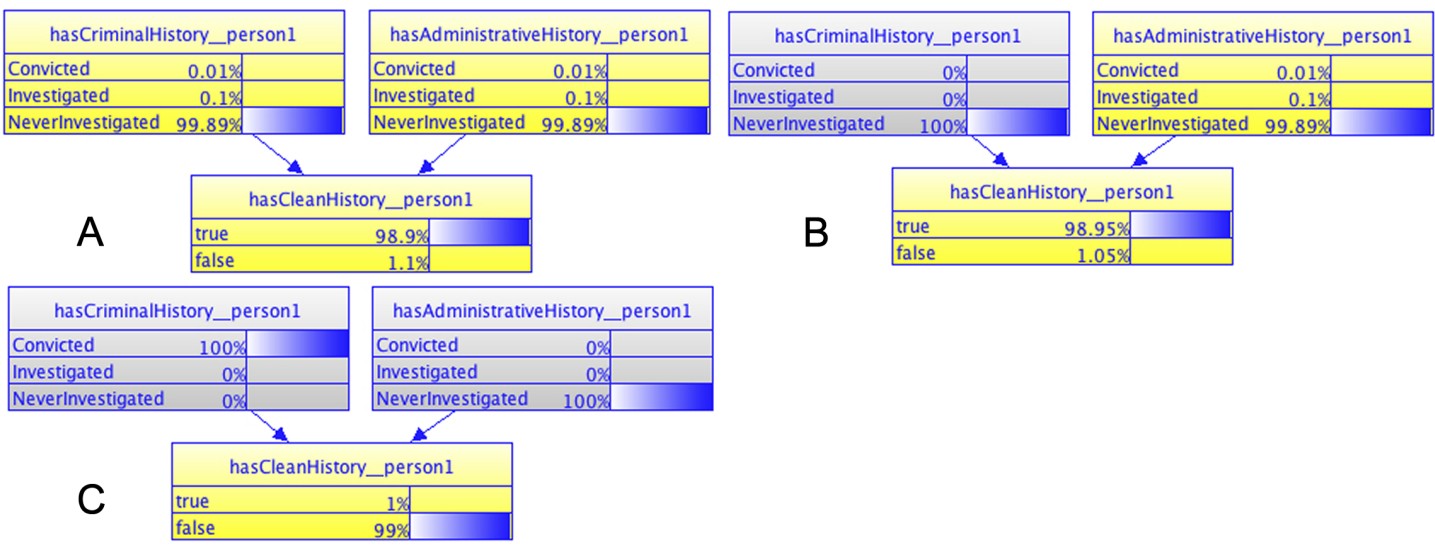

**Figure 9** Results of unit testing for the `Judgment History` MFrag.

results in a marginal "a priori" probability of 1.1% that any given person would not have a clean history. Figure 9B shows the model results when knowledge about NeverInvestigated is entered in the hasCriminalHistory_person1 RV (hasCriminalHistory__person1 node is colored in gray to show that it is observed). This causes a slight reduction in the belief that person 1 does not have a clean history (down from 1.1% to 1.05%). Finally, Fig. 9C shows the model's results when evidence on person 1 having a criminal conviction and never being investigated on an administrative process are entered (both parents now shown in gray with 100% probability on the observed value).

A systematic unit test would examine other combinations as well (*Carvalho, 2011*). It is important that unit testing achieve as much coverage as possible, and that results be analyzed by verifying that posterior probabilities behave as expected. In our case, the posterior probabilities are consistent with the expected result as defined by the expert.

Case-based evaluation is conducted by defining a range of different scenarios and examining the results produced by the system for each of the scenarios. Case-based evaluation is a system level test appropriate for integration testing. For our procurement PO, we define scenarios with evidence represented in different MFrags. This means that each query response will require instantiating multiple parts of the model, helping to validate how the model works as a whole. This validation is important to whether the model's global performance matches the specialist's knowledge.

It is important to try out different scenarios in order to capture the nuances of the model. In fact, it is a good practice to design the scenarios in order to cover the range of requirements the model must satisfy (*Wiegers, 2003*; *Sommerville, 2010*). Although it is impossible to cover every scenario we might encounter, we should aim for good coverage, and especially look for important "edge cases." A traceability matrix relating unit tests and case-based evaluation scenarios to MFrags is a useful tool to ensure that test scenarios have achieved sufficient coverage.

Keeping in mind the need to evaluate a range of requirements, we illustrate case-based evaluation with three qualitatively different scenarios. The first one concerns a regular procurement with no evidence to support the hypothesis of an improper procurement or committee. The second one has conflicting evidence in the sense that some supports the hypothesis of having an improper procurement or committee but some does not. Finally, in the third scenario there is overwhelming evidence supporting the hypothesis of an improper procurement or committee.

When defining a scenario, it is important to define the hypothesis being tested and what is the expected result, besides providing the evidence which will be used. Table 2 presents a comparison between all three scenarios. It can be seen that the difference between the first and the second scenarios is that member 1 was never investigated administratively in the first scenario, but was in the second. In the third scenario, however, besides having the evidence that member 1 was investigated, we also have the evidence that person 1 and 3 live at the same address and that person 2 lives at the same address as member 3.

In the first scenario, we expect that procurement will not be deemed improper since the members of the committee have never been investigated in either administrative or criminal instances and we have no relevant information about the owners of the enterprises participating in the procurement.

When the query is presented to the system, the needed MFrags are retrieved and instantiated for the entities relevant to the scenario, resulting in an SSBN that answers the query. Figure 10 shows part of the SSBN generated from scenario 1. Evidence includes the fact that member 2, who in this SSBN is part of the procurement process being assessed, has never been investigated in either an administrative process or in a criminal process. As expected, the probability of both `isImproperProcurement(procurement1) = true` and `isImproperCommittee(procurement1) = true` are low, 2.35% and 2.33%, respectively. In other words, the procurement is unlikely to be improper given the evidence entered so far.

In the second scenario, one of the three members of the committee was previously investigated in the administrative instance. All other evidence is the same as in the previous scenario. We expect that this new piece of evidence should not be strong enough to make the procurement improper, although the probability of being improper should be higher than in the first scenario.

The results of inference are as expected. The probability of `isImproperProcurement (procurement1) = true` and `isImproperCommittee(procurement1) = true` are 20.82% and 28.95%, respectively.[4] In other words, the probability increased but it is still relatively unlikely. However, depending on the stringency of the threshold, this case might be flagged as warranting additional attention.

Finally, in the third scenario, we have evidence that the owners of two different enterprises participating in the procurement process live at the same address. Since there are only three enterprises participating in the procurement, the competition requirement is compromised. Thus, the procurement is likely to be improper.

[4]The SSBN generated for this scenario is shown in *Carvalho (2011)*, provided as Supplemental Information.

Carvalho et al. (2016), *PeerJ Comput. Sci.*, DOI 10.7717/peerj-cs.77

**Table 2  Comparison of all three scenarios.**

| Scenario | Hypothesis and expected result | Evidence | Result |
|---|---|---|---|
| | | Evidence that apply to all scenarios:<br>hasCriminalHistory(member2) = NeverInvestigated | |
| 1 | Low probability that isImproperProcurement (procurement1) = true<br>Low probability that isImproperCommittee (procurement1) = true | hasProcurementOwner(procurement1) = agency1<br>isMemberOfCommittee(member1, procurement1) = true<br>isMemberOfCommittee(member2, procurement1) = true<br>isMemberOfCommittee(member3, procurement1) = true<br>isParticipantIn(enterprise1, procurement1) = true<br>isParticipantIn(enterprise2, procurement1) = true<br>isParticipantIn(enterprise3, procurement1) = true<br>isProcurementFinished(procurement1) = false<br>isResponsibleFor(person1, enterprise1) = true<br>isResponsibleFor(person2, enterprise2) = true<br>isResponsibleFor(person3, enterprise3) = true<br>Evidence unique to this scenario:<br>hasAdministrativeHistory(member1) = NeverInvestigated | 2.35% that isImproperProcurement (procurement1) = true<br>2.33% that isImproperCommittee (procurement1) = true |
| 2 | Probability that isImproperProcurement (procurement1) = true%, between 10% and 50%<br>Probability that isImproperCommittee (procurement1) = true%, between 10% and 50% | Evidence unique to this scenario:<br>hasAdministrativeHistory(member1) = Investigated | 20.82% that isImproperProcurement (procurement1) = true<br>28.95% that isImproperCommittee (procurement1) = true |
| 3 | Probability that isImproperProcurement (procurement1) =%, true greater than 50%<br>Probability that isImproperCommittee (procurement1) =%, true between 10% and 50% | Evidence unique to this scenario:<br>hasAdministrativeHistory(member1) = Investigated<br>livesAtSameAddress(person1, person3)<br>livesAtSameAddress(person2, member3) | 60.08% that isImproperProcurement (procurement1) = true<br>28.95% that isImproperCommittee (procurement1) = true |

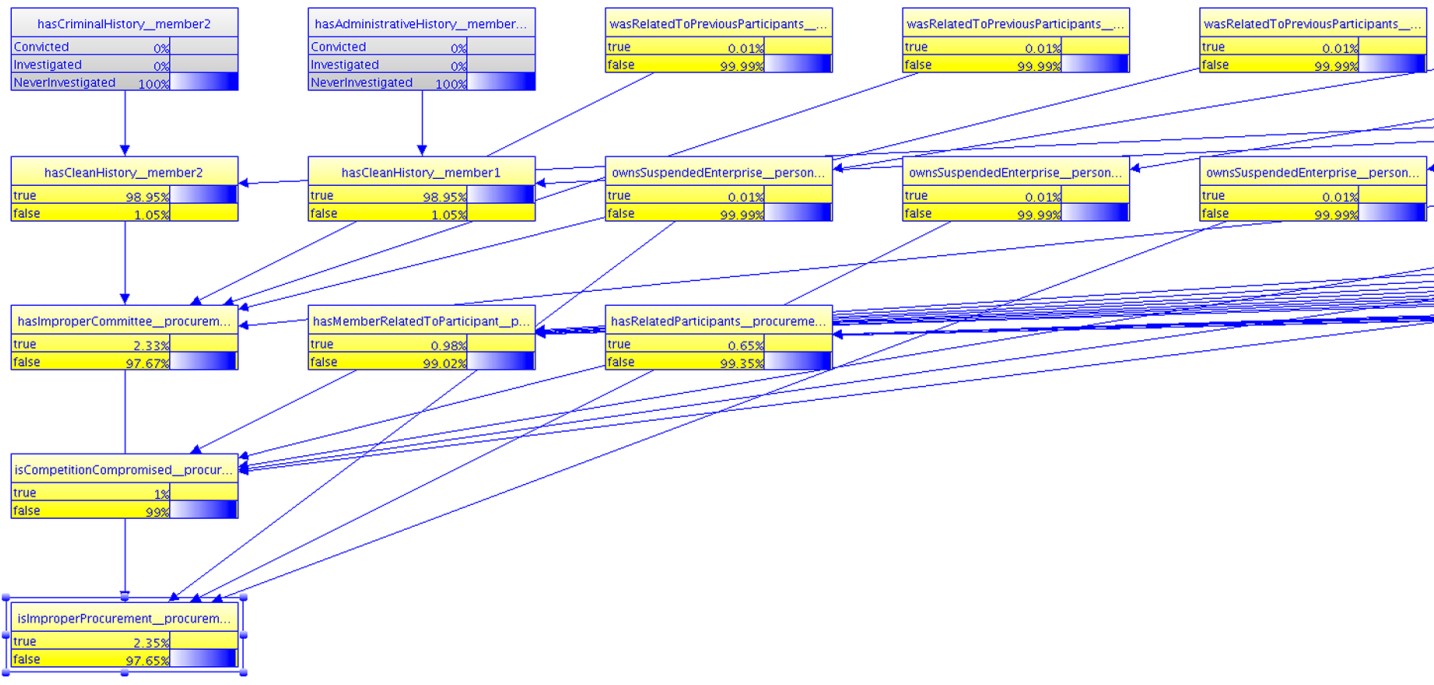

**Figure 10   Part of the SSBN generated for the first scenario.**

[5]The SSBN generated for this scenario is shown in *Carvalho (2011)*, provided as Supplemental Information.

As expected, the probability of `isImproperProcurement(procurement1) = true` and `isImproperCommittee(procurement1) = true` are much larger, at 60.08% and 28.95%, respectively.[5]  Notice that although the probability of having an improper procurement correctly increased to a value greater than 50%, the probability of having an improper committee has not changed, since there is no new evidence supporting this hypothesis.

The cases presented here are meant to illustrate the UMP-ST. A full case-based evaluation would consider a broad range of cases with good coverage of the intended use of the model.

## APPLICABILITY OF UMP-ST TO OTHER DOMAINS

In this paper, we focused on the fraud identification use case as a means to illustrate the core ideas of the UMP-ST. We chose this use case because its applicability was clear and its benefits have been independently tested (the methodology is currently being evaluated for use by the Brazilian Office of the Comptroller General). Nevertheless, the methodology is applicable to any problem requiring the development of a probabilistic ontology. Other examples of using the technique can be found in the terrorist identification domain (*Haberlin, Costa & Laskey, 2014*) and the maritime situation awareness (MDA) domain (*Haberlin, 2013*; *Carvalho et al., 2011*). For instance, the latter involved the development of a probabilistic ontology as part of the PROGNOS (Probabilistic OntoloGies for Net-centric Operation Systems) project (*Costa, Laskey & Chang, 2009*; *Carvalho et al., 2010*), in which PR-OWL was chosen as the ontology language due to its comprehensive treatment of uncertainty, use of a highly expressive first-order Bayesian language, and compatibility with OWL.

The MDA probabilistic ontology is designed for the problem of identifying whether a given vessel is a ship of interest. The probabilistic ontology was written in PR-OWL, and its development employed the UMP-ST process. An important aspect is that the development of the PR-OWL ontology was initially based on an existing ontology of Western European warships that identifies the major characteristics of each combatant class through the attributes of size, sensors, weapons, missions, and nationality. Thus, its development was a good case study for applying UMP-ST to extend an existing ontology to incorporate uncertainty.

During its development, the MDA ontology was evaluated for face validity with the help of semantic technology experts with knowledge of the maritime domain. This evaluation effort had issues in getting feedback from a sufficiently large number of experts, but the overall result of the evaluation suggests the UMP-ST not only as viable and applicable to the problem it supports but also a promising approach for using semantic technology in complex domains (*Haberlin, 2013*).

More recently, *Alencar (2015)* applied the UMP-ST process to create a PO for supporting the decision of whether or not to proceed with Live Data Forensic Acquisition. Besides using the UMP-ST, several tools and techniques shown in this paper were also applied: the use of UML Class Diagrams to identify the main entities, attributes, and relations for the model; the use of a traceability matrix to facilitate further improvements in the model; the implementation of the PO using PR-OWL and UnBBayes; and the validation of the model using both unit testing and case-based evaluation.

## FUTURE WORK

A natural next step in this research is the development of automated tools to support the UMP-ST. It is useful to have a tool to guide the user through the steps necessary to create a probabilistic ontology and link this documentation to its implementation in the UnBBayes PR-OWL plug-in. A tool to support this documentation process has been developed by the Group of Artificial Intelligence (GIA, Grupo de Inteligência Artificial, in Portuguese) from the Universidade de Brasília, Brazil (*De Souza, 2011*; *Santos et al., 2014*; *Carvalho et al., 2014*).

Penetration of semantic technology into serious applications cannot rely only on hand-engineered ontologies. There has been a robust literature on ontology learning (*Hazman, R. El-Beltagy & Rafea, 2011*) and learning of expressive probabilistic representations (*Getoor & Taskar, 2007*; *De Raedt, 1996*; *Luna, Revoredo & Cozman, 2010*), and there is robust research activity on the topic. The probability specification step of the POMC could combine both expert-specified probability distributions and probability distributions learned from data.

Practicing ontology engineers and the semantic technology community would benefit from widely available ontological engineering tools that support UMP-ST, provide scalable inference and support learning. In addition, there are other disciplines not discussed in this paper that are essential for practical ontology engineering, such as configuration management and user experience design. Further, the UMP-ST process would benefit

from a more detailed description of the activities performed, roles involved, and artifacts produced in its application.

The Eclipse Process Framework (EPF) could be employed to provide a structured way to present the disciplines, activities, best practices, roles, etc. As customizable software process engineering framework, EPF has two major goals (*Eclipse Foundation, 2011*):

- "To provide an extensible framework and exemplary tools for software process engineering - method and process authoring, library management, configuring and publishing a process."
- "To provide exemplary and extensible process content for a range of software development and management processes supporting iterative, agile, and incremental development, and applicable to a broad set of development platforms and applications".

Capturing UMP-ST within EPF will provide guidance and tools to a broad community of developers for following the UMP-ST to develop probabilistic ontologies. A process that is made freely available with the EPF framework is the OpenUP (*Balduino, 2007*) which is a minimally sufficient software development process. This process could be used as a starting point to describe the UMP-ST process, since OpenUP is extensible to be used as foundation on which process content can be added or tailored as needed.

Two major challenges must be addressed to enable broad use of semantically rich uncertainty management methods. The first is scalability. There have been some attempts to grapple with the inherent scalability challenges of reasoning with highly expressive probabilistic logics. For example, lifted inference (*Braz, Amir & Roth, 2007*) exploits repeated structure in a grounded model to avoid unnecessary repetition of computation. Approximate inference methods such as MC-SAT and lazy inference (*Domingos & Lowd, 2009*) have been applied to inference in Markov logic networks. Hypothesis management methods (*Haberlin, Costa & Laskey, 2010a*; *Haberlin, Costa & Laskey, 2010b*; *Laskey, Mahoney & Wright, 2001*) can help to control the complexity of the constructed ground network. Much work remains on developing scalable algorithms for particular classes of problems and integrating such algorithms into ontology engineering tools.

Finally, ontologies generated using the UMP-ST process would greatly benefit from methods that can assess how well and comprehensively the main aspects of uncertainty representation and reasoning are addressed. Thus, a natural path in further developing the UMP-ST is to leverage ongoing work in this area, such as the Uncertainty Representation and Reasoning Evaluation Framework (URREF) (*Costa et al., 2012*; *De Villiers et al., 2015*) developed by the International Society of Information Fusion's working group on Evaluation of Techniques for Uncertainty Reasoning (ETURWG). We are already participating in this effort and plan to leverage its results in the near future.

## CONCLUSION

The Uncertainty Modeling Process for Semantic Technology (UMP-ST) addresses an unmet need for a probabilistic ontology modeling methodology. While there is extensive literature on both probability elicitation and ontology engineering, these fields have developed nearly independently and there is little literature on how to bring them together

to define a semantically rich domain model that captures relevant uncertainties. Such expressive probabilistic representations are important for a wide range of domains. There is a robust literature emerging on languages for capturing the requisite knowledge. However, modelers can as yet find little guidance on how to build these kinds of semantically rich probabilistic models.

This paper provides such a methodology. UMP-ST was described and illustrated with a use case on identifying fraud in public procurement in Brazil. The use case was presented with a focus on illustrating the activities that must be executed within each discipline in the POMC cycle in the context of the fraud identification problem. The core concepts in applying UMP-ST to the procurement domain can easily be migrated to completely distinct domains. For instance, it was also used in defining a PO for Maritime Domain Awareness (MDA) (*Carvalho, 2011*) which supports the identification of terrorist threats and other suspicious activities in the maritime domain. The MDA PO evolved through several versions, showing how the UMP-ST process supports iterative model evolution and enhancement.

## ACKNOWLEDGEMENTS

The authors would like to thank the Brazilian Office of the Comptroller General for their active support since 2008 and for providing the human resources necessary to conduct this research. More specifically, the authors would like to thank the Department of Research and Strategic Information for providing the experts who explained the fraud domain in detail and provided help on creating and validating the use case described.

The authors would also like to thank Dr. Marcelo Ladeira and his students from the Universidade de Brasília for their work on developing the UnBBayes probabilistic network framework, and for providing support to this research as needed.

### Funding

This research was partially supported by the Office of Naval Research (ONR), under Contract #N00173-09-C-4008. The funders had no role in study design, data collection and analysis, decision to publish, or preparation of the manuscript.

### Grant Disclosures

The following grant information was disclosed by the authors:
Office of Naval Research (ONR): #N00173-09-C-4008.

### Competing Interests

Kathryn B. Laskey is an Academic Editor for PeerJ Computer Science. The authors declare there are no other competing interests.

## Author Contributions

- Rommel N. Carvalho conceived and designed the experiments, performed the experiments, analyzed the data, wrote the paper, prepared figures and/or tables, performed the computation work, reviewed drafts of the paper.
- Kathryn B. Laskey conceived and designed the experiments, analyzed the data, wrote the paper, prepared figures and/or tables, reviewed drafts of the paper, supervised software development, supervised research, directed Dr. Carvalho's PhD dissertation.
- Paulo C.G. Da Costa conceived and designed the experiments, analyzed the data, wrote the paper, reviewed drafts of the paper, supervised software development, served on Dr. Carvalho's PhD committee.

## Data Availability

The probabilistic ontology tool is available for download from http://sourceforge.net/projects/unbbayes/.

## Supplemental Information

Supplemental information for this article can be found online at http://dx.doi.org/10.7717/peerj-cs.77#supplemental-information.

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
