# Peer review of "Uncertainty modeling process for semantic technology"

_PeerJ Computer Science, doi:10.7717/peerj-cs.77_

## Round 0.1 · original submission · Major Revisions

· Academic Editor

Major Revisions

Please follow the recommendations of the reviewers closely since the paper needs to be revised significantly to be considered for publication. Especially more clarification in regard to cited methods and ontologies would improve the paper since a pure citation is not enough to introduce the readers to the methods but a comparison to the proposed method and the cited or applied ones, respectively, would clarify the relation between the presented and related work.

Also a minor point: Please don't use references in the abstract. The abstract should be fully self-contained.

Reviewer 1 ·

Basic reporting

1) The paper claims (in the abstract) that little has been done for the engineering of probabilistic ontologies. The statement is false based on industry, government, and the ETWURG. If you check the references and remove your own, the most recent paper is 2010? Clearly there is a mismatch between the paper preparation and the submission; or you choose not investigate the literature?
2) The use cases are too simple and not well vetted for applicability
3) The process should follow definitions in Figure 2 to make it easier for the reader to follow.
4) Math of uncertainty and the RV discussion is hard to discern how created, but is more associated with a discussion with BN. Some effort could be coordinated between a BN representation for the uncertainty quantification, that then is described in an ontology. The ontology then supports the query of analysis.

Experimental design

Too limited to discern viability

Validity of the findings

Does not seem reproducible of complete to validate

Additional comments

GRAMMAR and CONCEPTS
INTRO
1. “Uncertainty is fundamental across a wide range of domains to which semantic technology is applied”
-- awkward sentence
2. “This increase in complexity has been a major obstacle to penetration of expressive probabilistic representations into real-world applications.”
- awkward; the complexity is not a reason what it has not been reported in real-world literature (that comes from reporting on products).
3. “the literature contains little guidance on how to model a probabilistic ontology.” Seriously? Your references are 7-years old.
4. “use a probabilistic ontology in real world problems, the first step is creating and documenting the model (TBox).”
- what is the model – mathematical? Linguistic? Etc.
5. “t needs to be populated (A-Box).” - What is A, what is T?
6. “possible to query the model given the specific situation provided by the data available.”
- OK, a “query” is a way to access a system, but here you assume it is human or a machine
http://www.britannica.com/technology/query-language

UNCERTAINTY MODELING
7. Three stages? How can you build the model, without the parameters?
8. Uncertainty types – see the URREF
9. “Disciplines” – that is the word. Analysis and Design are separate concepts. In most engineering applications, you design first and then do the analysis of the design against testing. Not sure what the “unified process” is, but seems like it is myopically applied.
10. “in Figure 2, the requirements discipline (blue circle)” is labeled as Goals and not requirements. Do goals consist of queries and evidence? Seems like a disconnect? The other circisl do have some association with the terms.
12. Figure 2, LPD not described

FRAUD
13. What are the fraud types? All that is listed in “fake competition.” Where is a reference to a document that explores fraud?
14. “the prob ontology designed by experts transforms millions of items into dozens” – HOW?
15. 1. Goal: evidence – looks like a social network diagram
16. 2. Goal, query: evidence – question should be a “criminal history” extracted from a police database
17. RTM – you end with this, but it should be the start of the requirements section.
18. Figure 4 is good
19. Categories for development should follow Figure 2. Thus, there is no category “iimplementation” in Figure 2.
20. how does Figure 6 create the RV?
21. there is no example of the coordination of the math of the RV. For example “The Improper Procurement MFrag has the resident RV isImproperProcurement(procurement),” is not a RV.
22. If you have a LPD, you must have a GPD. It seems that you should look at the efforts for distributed fusion that works with graphs for distribution fusion.
23. Can you provide a easy to follow RTM matrix? There are no matrices presented
24. “The purpose of importance analysis is to verify that selected parts of the model behave as intended.”
This does not seem to follow as importance sampling is a decision process. You probable want to discuss “verification” that the model works as intended.
25. Too many references to Carvalho (2011) are required to learn from this presentation alone.
26. Need to highlight the scenario as a separate section as a use case. There is not enough detail to work through the examples as scenarios to follow the process. Numbers in Figure 9, 10 are not described in relation to the scenario.
27. The numbers for the scenarios are not compared to be able to learn from the examples. A combined table would be useful.

CONCLUSIONS
28. The quick look at the scenarios does not seem to convey the viability of the approach. More is needed to make the claim.
29. The second paragraph in the conclusions should go in the intro
30. “the next step” is part of future work.
31.”Future work” should be a discussion section before the conclusions.

Reviewer 2 ·

Basic reporting

The paper presents a methodology for building an ontology in the presence of uncertainty in the domain. The methodology is applied on a small subset of the Brasilian public procurements domain. The methodology is based on existing software development processes, encompassing requirements collection, analysis and design, implementation and test disciplines, across inception, elaboration, construction and transition phases.

The paper has as goal to fill the gap on modelling ontologies under uncertainty. However, I believe it is not clear in the paper what are the main differences concerning the modelling process when one has or has not an uncertain domain, i.e., besides making use of a language suitable for handling uncertainty, the paper should clearly present the specificities in the overall disciplines and phases that make possible to engineering an ontology representing uncertainty. Otherwise, the proposed methodology is the same as the ones handling traditional ontologies and what changes is only the implementation using a probabilistic ontology language.
Also, along the paper there are lots of citations to the previous work of the authors stating that more details and better explanation can be found there. I believe this harms the paper to be self-contained.
Thus, in a revised paper of the paper, regarding the overall method, I would like to see:
- how the uncertainty of the domain makes the phases and disciplines to be different from a standard ontology engineering modeling, as this is the gap that the paper proposes to fulfill; in addition what are the particularities of the UMP-ST process and the POMC compared to the traditional software development process.
- in case it is not possible to highlight how the overall process changes when facing uncertainty, I suggest the authors direct the paper to the development of a probabilistic ontology in the Brazilian procurement fraud domain. But, in this case, it is necessary to point out the contributions of the current paper compared to the previous ones by the same authors.
- the reasons for choosing the PR-OWL instead of others existing in the literature.
- how this paper contributes over the previous ones, and possibly related work concerning probabilistic ontology modeling (even if they are from the same authors)

Experimental design

The case study of the paper is a quite relevant domain, with the potential of helping in the decrease of corruption in Brazil. The UMP-ST is instantiated so that a probabilistic ontology can be built from this domain.
Some explanations about the domain are required, particularly, it should be clear in the paper what are the meaning of procurement, bidders, suppliers, etc, in the Brazilian context.
Following each discipline, please, try to answer the following issues in a revised version of the paper:
- in the current paper, only a small subset of the requirements is tackled, while the whole set is presented in the previous paper. Please, make clear how this subset was chosen, and how many requirements are there, and how this choice affects the overall process and the final developed probabilistic ontology, providing some examples.
- when defining the entities, its properties and relationships, the UML language is used, and the authors claim that this is because of the popularity of UML. However, I believe that nowadays OWL and its tools and editors are also quite popular among the ontology community. This also brings the necessity of specifying the rules separately, as UML has no support for that. Maybe there was a failed attempt of using OWL editors, but the popularity is not a hard claim for choosing UML, from my point of view.
- it is stated in the paper that everyone is obliged to file his/her TaxInfo, but this is a bit strong. It should be said that any person relevant to this domain are obliged to do that, not every Brazilian citizen.
- In page 8, in Implementation section, it is stated that PR-OWL2 is not currently implemented. At the end of this page, it is said that PR-OWL2 plugin is used. This is a little confusing. I believe this can be clearer if the authors simply explain the limitations of PW-OWL 1 in this section and the modifications that were introduced in order to attend such limitations. Then, PR-OWL2 can be left to the Future work section.
- Resident nodes, input nodes and the difference among them need to be explained in the paper. Actually, I believe the reader would benefit from some explicit background knowledge regarding this language.
- please, explain how the LPDs are obtained, whether they are manually defined or learned from data.
- the role of the expert should be more detailed in the Test section. Is there only one expert? How this choice can threaten the validity of the proof of concept?

Validity of the findings

My main concern related to the general findings of the paper is that the methodology is presented as general, however, this is not proved in the paper, as only one domain is taken into account and nothing is said about the people handling it. Not only the process should be presented but also a discussion about how following the POMC and UMP-ST helped or changed the previous way of modeling the domain. Moreover, it needs to be discussed in the paper the point of view of the experts, i.e., how a number of them would evaluate the proposed process, compared to following a different one, or no one. I believe this is essential to make the contribution of the paper really clear.

---

## Round 0.2 · Minor Revisions

· Academic Editor

Minor Revisions

Please address the final remaining comments for minor revisions of Reviewer 1.

Reviewer 1 ·

Basic reporting

The paper has a wealth of organization from previous work. There seems to be some style issues that are difficult. Figure 1 words on left side hard to read. Figure 2 does not work when printed in BW. There are keywords that need to be coordinated such as UP is not described (but assumed from the heading). The Brazilian translation is not described in the text on page 16 (DIE, CGU) which must be from Portuguese words and not the English.

1) Review section – might be better to italicize a few words in the descriptions as had to circle in detail to capture the organization – that is not possible from a quick read online

Experimental design

Excellent; however

2) The focus of the paper is the steps. This is still confusing. My guess is
Step 1) represent entities…
Step 2) determine uncertainty RV
Step 3) probabilistic distributions
Step 4) dependencies
Step 5) LPD
Step 6) define all LPDS
(Step 5 and 6 seem to be related)

Validity of the findings

REasonable

Additional comments

Grammar
1. Abstract: “is intended to support” implies one case, better “is demonstrated with an example to support”
2. Last line in “probability elucidation” – what is a prototype?
3. “and the UMP-ST is the focus” – of what? “this paper”
4. is it et al. in italics throughout or just when reused (Page 8)
5. Page 9 “requirements tracebility” – should be justified

Reviewer 2 ·

Basic reporting

The revised version of the paper is much clearer now, regarding both existing literature and the devised contribution. The main question that I have pointed out before, concerning the design of a probabilistic and deterministic ontology is better explained and exemplified in the proof of the concept.

Experimental design

The proof of concept presented in the paper has been thoroughly extended, with more details, figures, and explanations of the results.

Validity of the findings

The paper brings a proof of the concept that is now clear and useful enough for other people to write and experiment with its own probabilistic ontology, which as stated much clearer now in this revised version, is the main contribution of the paper.

---

## Round 0.3 · accepted · Accept

· Academic Editor

Accept

All the concerns of the reviewers have been addressed. The manuscript can be accepted for publication.